# Role of Akt/Protein Kinase B in Cancer Metastasis

**DOI:** 10.3390/biomedicines11113001

**Published:** 2023-11-08

**Authors:** Mohammad Islam, Sarah Jones, Ian Ellis

**Affiliations:** Unit of Cell and Molecular Biology, School of Dentistry, University of Dundee, Park Place, Dundee DD1 4HR, UK; s.j.jones@dundee.ac.uk (S.J.); i.r.ellis@dundee.ac.uk (I.E.)

**Keywords:** Akt, cancer, metastasis, HNSCC, EMT, cytoskeletal remodelling

## Abstract

Metastasis is a critical step in the process of carcinogenesis and a vast majority of cancer-related mortalities result from metastatic disease that is resistant to current therapies. Cell migration and invasion are the first steps of the metastasis process, which mainly occurs by two important biological mechanisms, i.e., cytoskeletal remodelling and epithelial to mesenchymal transition (EMT). Akt (also known as protein kinase B) is a central signalling molecule of the PI3K-Akt signalling pathway. Aberrant activation of this pathway has been identified in a wide range of cancers. Several studies have revealed that Akt actively engages with the migratory process in motile cells, including metastatic cancer cells. The downstream signalling mechanism of Akt in cell migration depends upon the tumour type, sites, and intracellular localisation of activated Akt. In this review, we focus on the role of Akt in the regulation of two events that control cell migration and invasion in various cancers including head and neck squamous cell carcinoma (HNSCC) and the status of PI3K-Akt pathway inhibitors in clinical trials in metastatic cancers.

## 1. Introduction

The primary reason for cancer-related deaths is metastatic disease [1]. The spreading of tumour cells from the primary lesion is the main cause for the mortality and morbidity of cancer patients, whether it exists at the time of diagnosis, progresses during treatment, or happens at the time of disease relapse [2]. The metastasis process involve a series of sequential, interconnected steps including: separation of tumour cells from the primary lesion and invasion of neighbouring, healthy connective tissue, intravasation into the blood and lymphatic vessels, circulation through the blood vessels (circulating tumour cells) to other tissues in the body, extravasation from the blood vessel into the new tissue, growth in specific distant organs, and building a secondary tumour [3,4,5] (Figure 1).

Recently, a novel ecological dispersal model of multidirectional cancer progression is proposed by Luo [6]. Taking nasopharyngeal cancer metastasis as an example, Luo hypothesized that the “nature of NPC is not a genetic disease but an ecological disease: A multidimensional spatiotemporal unity of ecological and evolution pathological ecosystem”. To adapt to the selective pressure from the remodelling microenvironment, NPC cells with cancer stem cells (CSCs) characteristics undergo EMT to dissociate from budding cells (tumour–host interface) and interplay with the local primary ecosystem (various stroma components); intravasate and survive into the circulation, and extravasate to circulating ecosystem (lymph node or a distant metastatic site); developing a distant metastatic ecosystem by entering slow-cycling states of dormancy, evading immune response, constructing organ-specific niches to colonise micro/macro metastases and later spread; self-feeding by CTCs or metastatic cells seeding at a distant site or secreting exosomes, cytokines, and chemokines and creating a multidirectional ecosystem by host cells including CAFs and immune cells to return to the primary tumour [6]. Many of the metastatic stages are dependent on tumour cell migration and invasion, which allows the cells to change tissue location. Tumour cells employ a similar mechanism of migration to spread to other tissues to those that happen in non-tumour cells during physiological events such as wound healing, angiogenesis, inflammatory immune responses, and embryonic morphogenesis [7]. However, tumour cell migration has been shown to be stimulated by diverse promigratory factors ignoring stop signals, including tumour cell-derived autocrine factors and the soluble factors present at secondary sites [8,9]. Due to this imbalance of signals, cancer cells become unceasingly migratory and invasive, causing tumour expansion across tissue boundaries and hence the development of cancer metastases [4,8].

Cell migration through tissues results from highly integrated multistep cellular events [7,10,11]. First, the moving cell polarises, elongates, and extends protrusions in the way of migration reacting to migration-promoting agents. There are two types of protrusions, which can be spike-like filopodia, or large and broad lamellipodia. Protrusions are typically guided by actin polymerisation and are stabilised by adhering to the extracellular matrix or adjacent cells via related transmembrane receptors [12]. Consequently, forward extension of a lamellipodium and retraction of the trailing edge causes the translocation of the cell body [8,12]. Reorganisation of the actin cytoskeleton is the most important processes of cell motility and is vital for most types of cell migration [13]. In the process of cell migration, the actin cytoskeleton is dynamically remodelled, and this reorganisation creates the physical force essential for cell migration [14].

Variable experimental behaviour and histological patterns of tumour cells suggest that tumour cells can utilise different cellular and molecular modes of migration based on cell-type-specific autonomous mechanisms and reactive mechanisms stimulated by the local microenvironments [15,16]. Tumour cells are detected as both single cells and organized collective sheets in malignant cancer patients, indicating that cancer cells exhibit the plasticity to switch between single and collective cell migration. Studies on single cell migration have founded the cellular and molecular basis, providing a significant understanding into the spreading of tumours whose cells migrate constitutively as single cells such as leukaemia or lymphomas, after separation from cohesive lesions through the epithelial to mesenchymal transition (EMT) [11,17]. Collective cell migration occurs when the junctions between cells are retained over extended periods of time, so cells are adherent to their neighbours. The efficiency of the metastatic process is increased by the transition to single cell migration. However, circulating grouped tumour cells detected in the patient peripheral blood samples suggests that the intravasation process can also be enacted by a cell cluster [18,19]. Cell migration is the first step to invasion. The extracellular matrix is degraded by invasive cells via proteolysis before entering neighbouring tissues [8,20].

Highly integrated multistep cellular events lead to cell migration and invasion through tissues that are regulated by various cell signalling pathways, including the PI3K-Akt signalling pathway. The serine/threonine kinase Akt is also known as protein kinase B (PKB). It was originally discovered as a proto-oncogene. Akt plays a significant regulatory role in various cellular activities including cell survival, cell migration and invasion progression, insulin metabolism, and protein synthesis and has thus become a focus of major attention. The Akt signalling pathway is activated by receptor tyrosine kinases (RTK), cytokine receptors, G-protein coupled receptors, integrins, B and T cells receptors, and other stimuli that stimulate the production of phosphatidylinositol 3,4,5, triphosphates (PIP3) through phosphoinositide 3-kinase (PI3K) [21]. The PI3 kinases are a set of lipid kinases that phosphorylate the membrane phospholipid, phosphatidylinositol 4,5 biphosphate (PIP2), generating phosphatidylinositol 3,4,5, triphosphates (PIP3). PIP3 controls a range of effector molecules including the Akt group of oncogenic kinases termed Akt1, Akt2, and Akt3. The activation of Akt1, a 60 kDa kinase, depends on PI3K [22]. An increase in cellular PIP3 by PI3K eventually allows the activation of Akt1 by phosphorylation at Thr308 and Ser473 residues [23]. This activation is completed by structural modification stimulated by PI3K-dependent kinase-1 (PDK-1)-dependent phosphorylation at Thr308 and stabilisation by mTORC2 or DNA-PK (DNA-activated protein kinase) dependent phosphorylation at Ser473 [24,25,26]. A third phosphorylation site on Akt1 has been identified at Thr450 [27]. This site is referred to as the turn phosphorylation site and is controlled by mTORC2 activity [28,29]. The activation of the three Akt isoforms plays a pivotal role in fundamental cellular functions such as protein synthesis, cell survival, proliferation, and autophagy by regulating a variety of downstream substrates such as mTORC1, MDM2, Cyclin D1, and Beclin1, respectively [21,30,31] (Figure 2).

There are frequent alterations of the PI3K-Akt pathway in various types of human cancers. Amplification of the PIK3C gene encoding PI3K or the Akt gene lead to the constitutive activation of the PI3K-Akt pathway. PTEN (phosphatase and tensin homologue deleted on chromosome 10) can inhibit the Akt activation, and mutation in the PTEN gene also causes the constitutive activation of Akt [32,33,34]. Recent evidence has also suggested that Akt plays an important role in cancer cell migration and invasion [35,36]. This review focuses on the regulatory roles of Akt in cancer cell metastasis including head and neck cancer, emphasising cell migration. This review also briefly updates the status of clinical trials with PI3K-Akt inhibitors alone or in combination therapy in metastatic cancers.

## 2. Akt in Cytoskeletal Rearrangements

The cytoskeleton is the supporting structure of cells which is composed of a filamentous network of micro filaments such as actin and myosin, intermediate filaments such as vimentin and keratin, and microtubules such as tubulin [37]. The main purpose of the cytoskeleton is to maintain cellular structure, intracellular transport, and supporting cell division. Cytoskeletal rearrangements occur in various physiological and pathological events such as cell movement, wound healing, and cancer metastasis [38]. Cellular motility either in physiological events or in pathological conditions is driven by cytoskeletal remodelling, initiated by various signalling pathways. The synergistic effect of all the three basic elements—filamentous actin, microtubules, and the intermediate filament vimentin—is the potential basis for a cell to migrate [35]. Wide-ranging studies have focused on how the stabilisation of intracellular filaments and dynamic polymerisation control cell migration [14,39]. Akt can phosphorylate a diverse group of key factors associated with the skeletal filaments.

Growth of the vascular network is essential for the spread of cancer cells. Angiogenesis is the process whereby new vessels are formed and involved in the supply of nutrients, oxygen, and immune cells and also the removal of waste products [40]. Angiogenic factors play a huge role in neoplastic vascularisation, thus increasingly gaining attention. Vascular endothelial cell migration is a vital step for angiogenesis. Vascular endothelial growth factor (VEGF) activates Akt and stimulates the migration of endothelial cells by increasing actin polymerisation. Abrogated Akt activity by expression of a kinase-dead mutant inhibits actin bundle formation and blocks cell migration. This effect is enhanced when myristylated Akt is expressed [41], demonstrating that Akt is a critical mediator of VEGF-induced endothelial cell migration through actin reorganisation. Data also suggest that eNOS activation via phosphorylation of Ser-1177 by Akt is necessary and sufficient for VEGF-mediated EC migration [42,43].

In chicken embryonic fibroblasts (CEF), PI3K-transduced migratory signal was blocked by inhibiting Akt activity. PI3K also activated p70S6K1 via Rac and induced actin filament remodelling and cell migration in CEF cells. This study confirms that the activation of PI3K activity alone is adequate to remodel actin filaments to increase cell migration through the activation of Akt and p70S6K1 in CEF cells [44]. Another study suggested that overexpression of the integrin-linked kinase (ILK) pathway is sufficient to stimulate PI3K-dependent Rac1 activation. Blocking of Akt, p70S6K1, or Rac1 inhibited the effect of ILK on actin filaments, hence blocking cell migration, implying a regulatory role for the PI3K/Akt/p70S6K1/Rac1 signalling pathway in response to ILK [45]. In ovarian cancer, p70S6K1, downstream of the PI3K/Akt pathway, stimulated the rapid activation of Rac1 and cdc42 and their downstream effector molecule p21 activated kinase (PAK1) [46]. In neutrophils, activation of G-protein coupled receptors results in F-actin polymerisation and cytoskeleton contraction through PIP3 signalling. This pattern of actin reorganisation ensures pseudopod extension in human neutrophils during chemoattractant stimulation, which is dependent on Akt activity [47]. Breast cancer cell migration and invasion often occurs in an Akt dependent manner which is characterised by increased filopodia production. A specific Akt inhibitor named API-2 (Akt phosphorylation inhibitor 2) blocks breast cancer cell migration by blocking filopodia formation [48]. These observations of Akt activation and its role suggest that Akt can potentially regulate cell migration through direct modulation of actin.

Other studies have shown that actin preferentially binds to phosphorylated Akt at pseudopodia with enriched bundles [49,50]. Another study further confirmed that Akt can phosphorylate actin and therefore cortical reorganisation of actin associated with cell migration is strongly dependent on Akt activation [51]. Studies with HeLa cells revealed that Akt phosphorylates PAK1, a protein which belongs to the p21-activated serine/threonine kinase family and facilitates its binding with the non-catalytic region of the tyrosine kinase adaptor protein (Nck) promoting chemotaxis [52]. This effect of Akt through PAK1 may be mediated by enhanced myosin 2 assembly and polarity [53].

The actin-rich structure of highly motile cells like invadopodia, filopodia, and pseudopodia needs to be stabilised to function properly. Actin-associated proteins are responsible for stabilising this actin structure by blocking the degradation of newly formed actin filaments [54]. ALE (the Akt phosphorylation enhancer), also termed the ‘girder’ of actin filaments (Girdin), is one of the best examples of this type of protein. APE/Girdin provides the integrity of the actin meshwork (actin filament) at the leading edge of migrating cells. Reduction in APE/Girdin destabilises the actin bundles, triggering the ablation of stress fibres and actin structure. This results in the loss of directional migratory ability and establishes the vital activity of APE/Girdin in the regulation of cell migration. Enomoto et al. proved that APE/Girdin is phosphorylated by Akt on Serine 1416 (S1416) [55]. Upon stimulation by EGF, S1416 phosphorylation initiates the translocation of APE/Girdin, regulating actin reconstructions and Akt-controlled cell motility in cancer-associated fibroblasts, fibroblasts, breast cancer, and oesophageal squamous cell carcinoma cells [56,57,58,59,60]. Akt has also been shown to promote actin reorganisation and cell motility mediated by the mechano-protein and Akt substrate ANKRD2 (Ankyrin repeat domain protein 2, also known as ARPP) [61].

An actin-associated structural (cross-linker) protein, filamin A, is phosphorylated by Akt on residue S2152 [62,63,64]. In turn, phosphorylated filamin A mediates caveolin-1-induced cancer cell migration through the IGF signalling pathway [65,66]. Akt has been shown to phosphorylate NHE1 (sodium-hydrogen exchanger isoform 1), a key mediator of stress fibre disassembly on S648 and suggested to be critical for the growth factor-induced cytoskeletal rearrangements that favour cell migration and invasion [67]. Other studies have demonstrated the migration of different cell types by modulation of the cytoskeleton through NHE1, although the role of Akt was not elucidated [68,69,70,71]. A study in fibroblasts demonstrated that the Akt pathway is necessary for the translocation of NHE1 to the leading edge and actin nucleation at the lamellipodium that supports directional cell migration [72].

Extensive studies have been carried out to investigate the role of intermediate filaments in cell motility [73,74]. The most abundant intermediate protein that maintains normal cell and tissue integrity is called vimentin, a type three filamentous protein. It is phosphorylated by Akt1 on residue S39, stabilised, and thereby regulates cancer cell invasion in aggressive sarcoma [75]. It has also been shown that vimentin is highly expressed in breast cancer lung metastases [76,77]; however, the specific mechanisms to control cell migration by some Akt substrates are still undefined. For example, S-phase kinase-associated protein 2 (skp2), a component of E3 ligase, is phosphorylated by Akt on the S72 residue, stimulates Skp-2 dependent ligase activity, and induces cell migration [78,79]. Akt also promotes cell migration by regulating microtubule dynamics through Akt/GSK3 beta axis-dependent activation of the microtubule binding protein, APC (adenomatous polyposis coli) [80,81,82].

Akt interacts with promigratory proteins, in addition to targeting cytoskeletal proteins, thus facilitating crosstalk between associated signalling axes. The VEGFR/eNOS signalling pathway-controlled cell migration is dependent on Akt-mediated phosphorylation on S1177 [42]. Accumulating evidence has indicated the importance of nitric oxide (NO) in pathological conditions, especially in malignant tumours [83,84]. Furthermore, VEGFR signalling often cooperates with the G-protein coupled receptor, sphingosine-1-phosphate receptor 1 (SIPR1, also known as endothelial differentiation gene 1, EDG-1). SIP/SIPR1 activation leads to the phosphor-activation of VEGFR which phosphorylates Src kinase, consequently activating the PI3K/Akt/eNOS axis [85]. Akt-mediated phosphorylation of SIPR1 on T236 further enhances their activity and stimulates cortical actin assembly, angiogenesis, and chemotaxis [86,87]. Thus, Akt plays a vital role in regulating VEGFR and the SIP/SIPR1 signalling pathway and actively regulates cell migration. EphA2 (Ephrin receptor tyrosine kinase A2), a member of the largest tyrosine kinase family, is also phosphorylated by Akt on S897 residue. In human brain cancer cells, S897 phosphorylation in EphA2 is responsible for cell migration and invasion through dendritic actin cytoskeletal rearrangements and lamellipodia formation [88,89]. Scientists have shown that EphA2 recruits Ephexin 4 (a guanine nucleotide exchange factor for the small GTPase, RhoG) upon phosphorylation of S896 and promotes breast and colorectal cancer cell migration and anoikis resistance [90].

It is now well established that membrane redistribution of integrin by various signalling pathways is a critical mediator of cellular movement. The ANK repeat and pleckstrin homology domain-containing protein 1 (ACAP 1) is a GTPase activating protein (GAP) for ADP ribosylation factor 6 (ARF6) known to participate in integrin beta recycling. ACAP1 is phosphorylated by Akt on S554, stimulates integrin recycling, and therefore promotes cell migration [91]. Another GTPase activation protein, RhoGAP22, is shown to be phosphorylated by Akt on S16, upon stimulation by insulin or possibly PDGF, and this plays a vital role in regulating cell migration, leading to modulation of Rac1 activity [92]. Various studies have established the role of the mammalian targets of rapamycin complex 1 (mTORC1) in the cell migration and relationship with Akt [93,94]. Akt regulates mTORC1 activation and, in turn, activates the phosphorylation of p70S6K1 (S6K1) and inhibits eukaryotic initiation factor 4E binding protein (4EBP1). It is suggested that tuberous sclerosis complex 1/2 (TSC 1/2), a tumour suppressor gene, inhibits S6K1 and activates 4EBP1, facilitated by inhibiting mTORC1. Akt-mediated phosphorylation of TSC2 also destabilises the complex and activates mTORC1 [95]. In the single cell motility assay, IGF-1-stimulated cell motility was inhibited by downregulation of S6K1 using lentiviral and ectopic expression of constitutively hypophosphorylated 4EBP1 [96]. S6K1 regulates cell motility, which might be related to regulating phosphorylation of focal adhesion kinase (FAK), paxillin, p130^cas^, and F-actin organisation (or lamellipodia formation) [97]. Furthermore, mTORC1 mediates phosphorylation of ERK1/2 (extracellular signal related kinase) on T202 through direct and indirect regulation of PP2A (protein phosphatase 2A). Inhibition of PP2A activates ERK1/2 and promotes motility in several transformed cancer cells [98,99,100,101,102].

Several studies have also demonstrated that transforming growth factor beta 1 (TGFβ1) enhances human chondrosarcoma and lung cancer cells’ migration through the PI3K/Akt signalling pathway. Akt phosphorylates IKKαβ (IkB kinase) which activates IkBα and p65 on S536 residue. This causes NFkB to dissociate from IkBα and hence activate β1 and αvβ3 integrin, promoting human lung cancer and chondrosarcoma cell migration [103,104]. Abrogation of mTOR signalling leads to the lack of functional mTORC1 in human trophoblast cells. mTORC1 regulates the JAK/STAT signalling pathway and contributes to the invasiveness of trophoblast cells by regulating matrix remodelling enzymes such as MMP9 (matrix metalloproteinase), MMP2, uPA (urokinase plasminogen activator), and PAI-1 (plasminogen activator inhibitor) [9,105]. The opposing role of Akt in cell migration has also been discussed in different studies. Akt phosphorylates kidney ankyrin repeat-containing protein (Kank), which consequently leads to a negative regulation of stress fibre assembly and RhoA activation, attenuating cell migration [106]. An actin binding protein, paladin, phosphorylated by Akt1 on S507, inhibits breast cancer cell migration by disrupting F-actin bundles [107]. On the other hand, Akt2 contributes to paladin stability independent of S507 phosphorylation [108]. Similarly, Akt phosphorylates TSC2 (tuberous sclerosis complex), a Rho GTPase regulator that inhibits breast cancer cell migration due to impaired F-actin assembly [109].

## 3. Akt in EMT

Epithelial cells are tightly connected to their adjacent cells via E-cadherin and with actin filaments via α- or β-catenin. Epithelial tumour cells must break these intercellular junctions before migrating as single cells and invading stromal tissues. Epithelial tumour cells undergo a process named epithelial to mesenchymal transition (EMT) to facilitate the invasion as a single cell. The EMT process can be induced either by extracellular growth factors, for example EGF, TGF-α and β, FGF, or by intracellular cues, such as oncogenic Ras [110,111]. Epithelial cells gain a mesenchymal phenotype by losing their polarity and cell–cell contacts during EMT. Functional loss of E-Cadherin and downregulation of epithelial cell markers such as cytokeratins and ZO-1, and the overexpression of mesenchymal or fibroblast cell markers such as N-cadherin, vimentin, and fibronectin are the main characteristics of EMT [112,113]. EMT is a complex biological process that plays a critical role in cancer metastasis. In head and neck cancer, EMT can be involved in the dissemination of cancer cells to distant sites. However, it is important to understand that EMT is not an all-or-nothing phenomenon; there are partial or hybrid states of EMT that can have unique implications for cancer metastasis.

Partial EMT (p-EMT) is a term used to describe a state in which cancer cells exhibit some, but not all, of the characteristics associated with a full EMT [114]. p-EMT can enhance the invasive capacity of cancer cells. These cells may have an increased ability to break away from the primary tumour as a group of cells and infiltrate surrounding tissues, which is a crucial step in metastasis. Partially EMT-activated cancer cells might be less susceptible to apoptosis. This allows them to survive in the bloodstream and at distant site, where they might otherwise be eliminated by the body’s natural defences [115,116]. Cells which undergo p-EMT may also evade the immune system to some extent, making it more challenging for the body to recognize and destroy these cells. Partial EMT can also contribute to the formation of a pre-metastatic niche at distant sites. This involves the recruitment and conditioning of stromal cells and immune cells to create a supportive microenvironment for incoming cancer cells [117,118]. Puram et al. (2017) provided evidence that TGFβ signals from CAFs in the stroma induced p-EMT at the leading edge of HNSCC tumours by upregulating Snail2 expression, potentially promoting invasive properties of this subpopulation [119] (Figure 3).

EMT is reversible and, sometimes, cells undergo the reciprocal mesenchymal to epithelial transition (MET). During the development process, EMT plays an essential role in the development of various tissues and organs such as the heart, neural crest, and peripheral nervous and musculoskeletal systems. Only a small number of cells in adult organisms have the ability to go through the EMT process in specific physiological or pathological events such as wound healing. Nevertheless, tumour cells often gain the ability to reactivate the EMT process during metastasis, which enhances the migration and invasion capacity of cancer cells [113,120]. A number of studies have reported that Akt is frequently activated in human carcinomas [121,122,123,124,125]. Akt2 has been shown to be activated in ovarian carcinoma, affecting epithelial cell morphology, tumorigenicity, cell motility, and invasiveness, which is characterised by the loss of histological features of epithelial differentiation [126]. Evidence that Akt was shown to regulateEMT was first published in 2003, where squamous cell carcinoma cells, overexpressing an activated mutant of Akt, were shown to undergo EMT and downregulate E-cadherin [127]. Loss of E-cadherin and relocalisation of β-catenin from the membrane to the nucleus is frequently detected in tumour cells undergoing EMT [128,129]. Several transcription factors have been recognized that can induce and maintain the EMT process, including Snail, Twist, and Zeb. The definitive molecular signalling mechanisms of normal regulation of these transcription factors are still uncertain; however, they are apparently deregulated in many invasive cancers [112,130]. Evidence suggests a strong relationship between Akt and EMT-inducing transcription factors. Snail is phosphorylated by GSK3β (glycogen synthase kinase 3 beta) in normal epithelial cells but is very unstable and hardly detectable. Expression of Snail in epithelial cells strongly induces morphological changes associated with enhanced migratory capacity [131,132]. Phosphorylated Akt downregulates GSK3β by phosphorylating the S9 residue. GSK3β activates β-catenin and Snail that leads to their ubiquitination and degradation. Abrogated GSK3β, on the other hand, causes the stabilisation and nuclear accumulation of β-catenin and Snail. Nuclear Snail suppresses transcription of the CDH1 gene encoding E-cadherin to stimulate the EMT process. Abrogated GSK3β stabilised the transcription factor Snail and increases the expression of vimentin, N-cadherin, and MMP-9. Nuclear β-catenin stimulates the transcription of cMYC and the cyclin D1 gene, which plays a vital role in the EMT process. This is possibly consistent with invasive cancers, where increased Akt phosphorylation leads to the downregulation of GSK3β and Snail overexpression [133,134,135,136]. A recent study also suggested that the activation of the Akt/GSK3β/Snail pathway induced by Collagen type X1 α1 (COL11A1) plays a major role in the progression of pancreatic ductal cancer by facilitating EMT [137].

Y-box binding protein-1 (YB-1), a transcription/translation regulatory protein, is reported to be activated by Akt and translocated to the nucleus. Nuclear YB-1 thus phosphorylates Snail and decreases E-cadherin expression, which in turn induces EMT in invasive breast carcinoma [138]. Furthermore, upregulated Snail could, in turn, increase Akt activity. Snail increases the binding of Akt2 to the E-cadherin (CDH1) promoter and Akt2 interference unexpectedly inhibits Snail repression of the CDH1 gene [139]. Akt2 could also be activated by another EMT-inducer, Twist, in invasive breast cancer cells [140]. Inhibition of Akt also downregulates Twist in cancer cells [80]. Furthermore, Akt phosphorylates and activates Twist1, which in turn enhances the phosphorylation of Akt because of increased TGFβ signalling in human breast cancer [141,142,143]. Data also suggest that the polycomb group protein named B lymphoma Mo-MLV insertion region 1 homolog (Bmi1) is a downstream target of Twist1 and is crucial for EMT and cancer metastasis [144]. Akt can phosphorylate Bmi1 directly in high-grade prostate tumours [145]. Promotion of Akt activity by Bmi1 was also found to promote EMT by blocking the GSK3β-mediated degradation of Snail in HNSCC and breast cancer [146,147]. Twist and Bmi1 also mediate suppression of a micro-RNA, miR let-7i, which results in NEDD9 and DOCK3 overexpression and promotes mesenchymal motility in HNSCC, melanoma, and breast cancer via Rac1 [148,149,150]. In many cases, breast cancer metastasis may be under the control of balance between Akt1 and Akt2 and their link with MiR-200/Zeb/E-cadherin axis [151,152]. Taken together, numerous studies establish the significant interaction between Akt and EMT inducer-associated signalling. This synergistic interaction has serious adverse pathological effects: (1) it sustains upregulation of PI3K/Akt signalling, which increases further the anti-apoptotic potential of cancer cells; (2) it induces pro-invasive/metastatic gene expression; and (3) it halts the stress-induced cell cycle arrest in cancer cells [35,113,134]. It is essential to note that partial EMT is a dynamic and heterogeneous process, meaning that not all cancer cells within a tumour will exhibit the same characteristics or degree of EMT. The presence of partial EMT within a tumour can complicate treatment strategies, as these cells may respond differently to therapies [114].

Figure 4 illustrates the role of Akt in regulating downstream signalling molecules that in turn regulate cytoskeletal remodelling and EMT events in cancer cells.

## 4. Akt in HNSCC Metastasis

Head and neck squamous cell carcinoma (HNSCC) denotes epithelial tumours that develop in the oral cavity, pharynx, larynx, and nasal cavity. The main risk factors of HNSCC are alcohol and tobacco use and HPV infection [153,154]. It is the seventh most common cancer worldwide, with more than 887,000 cases and 450,000 deaths every year (accumulation of different head and neck cancer sites) [155]. It has recently been shown that Akt is persistently activated in the vast majority of HNSCC cases. Active forms of Akt (phosphorylated) can readily be detected in both experimental and human HNSCCs and HNSCC-derived cell lines [156,157,158]. Akt can be phosphorylated, hence activated by different growth factors, chemokines, integrins, etc., and their respective receptors, ras mutations, PI3Ka gene amplification, overexpression, or activating mutations. Akt can also be activated by aberrant PTEN activity due to genetic alterations or reduced expression in HNSCC [159,160]. Akt activation is an early event in HNSCC progression which can be identified in as many as 50% of tongue preneoplastic lesions [161]. Akt activation also represents an independent prognostic marker of poor clinical outcome in both tongue and oropharyngeal HNSCCs [161,162] and is linked with the conversion of a potentially malignant oral lesion to OSCC (oral squamous cell carcinoma) [163].

Akt is known to induce morphological changes associated with EMT, loss of cell–cell adhesion, and increased motility and invasion in HNSCC [112]. Oral carcinoma cells, of epithelial origin, ectopically express a mesenchyme-specific transcription factor (HMGA2) at the invasive front, which has a significant impact on tumour progression and patient survival [164]. However, the definitive evidence that EMT was induced by Akt was provided by a study in which oral squamous cell carcinoma cell lines overexpressing activated mutant Akt were shown to undergo EMT and downregulate E-cadherin [127]. Snail and SIP1 exhibit an inverse correlation with E-cadherin expression levels in oral carcinoma cells [165,166]. An OSCC clone with stable Snail overexpression displayed spindle morphology, amplified expression of vimentin, and reduced expression of E-cadherin [167]. Julien et al. reported that phosphorylation of NF-κB by Akt stimulates Snail expression and induces EMT in OSCC [168]. Bmi1 was found to bind with the promoter of the Akt inhibitor, PTEN and thus promoted Akt activity and in turn EMT by blocking GSK3β-mediated degradation of Snail. Interestingly, Bmi1 binds to the E-cadherin promoter but depends on Snail for E-cadherin repression. Thus, Bmi1 was found to be a player in EMT by activation of Akt, stabilisation of Snail, and repression of E-cadherin in HNSCC [134,147]. Increased Twist expression is associated with downregulation of E-cadherin and may also influence the Akt pathway through an unclear mechanism in nasopharyngeal carcinoma cells [169]. Another study showed that pAkt inhibition could induce mesenchymal to epithelial transition (MET) though interaction between Twist and pAkt during EMT in OSCC [80]. The SDF-1/CXCR4 system can also induce EMT via activation of the PI3k-Akt signalling pathway, resulting in lymph node metastasis of OSCC [170]. NOTCH1-inactivating mutations are observed in around 30% of HNSCC cases which activate cell proliferation and EMT though the induction of the EGFR/PI3K/Akt axis [171,172].

Research from our group suggested that VEGFA stimulated OSCC and oral cancer- associated fibroblast cell migration and can be inhibited by a specific PI3 kinase and mTORC2 inhibitor. Addition of VEGF also caused increased Akt phosphorylation at both T308 and S473 residues. The phosphorylation of Akt was found to vary according to the concentration of VEGF, cell types, incubation time, and assay format [157]. Although it has been suggested that differential phosphorylation of Akt at these two sites may modulate the substrate selectivity of Akt, a clear picture of this is yet to emerge [24]. In another study, we also found the nuclear localisation of pAkt T308 both in VEGF-induced migrated oral carcinoma cells and VEGF-positive head and neck cancer tissue, while pAkt S473 was mainly localised in the cytoplasm. Vasco et al. showed that the localisation of phosphorylated Akt varies between two forms of thyroid cancer, but nuclear localisation is linked with tumour invasion in both subtypes [173]. Akt has been reported to be abundant in the nucleus in many cancer cells, yet the mechanism of translocation, biological importance, and activity has not yet been established [174]. Published data from our group also revealed that EGF (epidermal growth factor), TGFα (transforming growth factor α), TGFβ1 (transforming growth factor β1), and NGF (nerve growth factor) can stimulate head and neck cancer cell migration, and a specific PI3k/Akt pathway inhibitor such as PI103 or MK2206 can effectively block growth factor-induced cell migration [175,176]. A study from our research group also suggested that receptor tyrosine kinase inhibitors such as Gefitinib and Erlotinib inhibited the migration of head and neck cancer cells by inhibiting both Akt and MAPK phosphorylation [177]. Cetuximab, a monoclonal antibody, targeting EGFR is the only FDA-approved targeted therapy for the treatment of recurrent/metastatic head and neck cancer, in combination with radiation therapy or as a single agent in patients who have had prior platinum-based therapy. The response rate, as a single agent, is only 13% and the patients who respond initially eventually develop resistance [178]. Evidence showed that an EGFR mutation at S493R inhibits Cetuximab-binding with the receptor but does not block EGF or TGFα binding. EGF or TGFα may therefore activate the downstream PI3K/Akt signalling pathway. Cetuximab resistance can also be mediated by the activation of the Akt signalling pathway in an alternative way, such as the overexpression of other growth factors (TGFβ, VEGF, and NGF) and their associated receptors by the tumour cells and/or the tumour microenvironment [179]. A recent study also demonstrated increased Akt 1/2/3 phosphorylation to be the cause for acquired Cetuximab resistance in head and neck squamous cell carcinoma [180]. The PI3K/Akt pathway can also promote tumour immune evasion by modulating the tumour microenvironment, upregulating immunosuppressive molecules, and inhibiting the activity of cytotoxic T cells. Inhibiting this pathway by blocking immune checkpoint molecules, PD-1 or PD-L1, may help to sensitise HNSCC tumours to immunotherapy by creating a more favourable immune microenvironment. A potential treatment strategy might involve the inhibition of PI3K/Akt pathway which can help reduce tumour growth and sensitise cancer cells to immune attack and administer an immune checkpoint inhibitor concurrently, to recognize and attack cancer cells more effectively. There is some preclinical and early clinical evidence suggesting that the combination of PI3K/Akt inhibitors and immunotherapy agents such as PD-1/PD-L1 inhibitors may have synergistic effects, resulting in improved anti-tumour responses [181,182].

The growth of HNSCC is maintained by a population of specialized cells, cancer stem cells (CSCs), which possess unlimited self-renewal potential and induce tumour regrowth, if not eliminated by therapy. Given their self-renewal properties, CSCs are thought to play a key role in tumour growth and metastasis, but also in recurrence, making CSC-related gene and protein expression a promising biomarker candidate and therapeutic target [183]. Evidence suggests that HNSCC metastasis is associated with Bmi1-positive CSCs, which are responsible for tumour invasion, drug resistance, and lymph node metastasis. Migration/invasion abilities, cancer stemness and EMT phenotype of HNSCC CSCs are maintained by the Twist/Bmi1/Akt/β-catenin signalling pathway [184,185]. Thus, targeting the Akt pathway in HNSCC CSCs could be an innovative way to treat cancer whilst avoiding drug resistance.

## 5. PI3K/Akt Inhibitors in Clinical Trials

A few recent clinical trials using PI3K/Akt inhibitor alone or in combination with other agents to treat metastatic cancer did not show promising outcomes (Table 1).

It is worth noting here that activated receptor tyrosine kinases activate not only the PI3K-Akt signalling pathway, but also other pathways including the MAPK and SMAD pathways. Signalling pathways are activated in a context-dependent manner and crosstalk among each other. Hence, targeted inhibition of one pathway downstream of the receptors may not affect other pathways and that adds complexity to therapeutic targeting. A recent study suggested that the combination of an Akt inhibitor and Cetuximab might be a favourable novel therapeutic strategy to overcome acquired Cetuximab resistance in HNSCC patients [180]. Patient selection, tumour characteristics, and the specific agents used can also influence the effectiveness of the combination therapy.

## 6. Conclusions

Extensive studies have demonstrated that the activation of Akt by phosphorylation of different amino acid residues determines substrate selectivity and thus exerts different biological activity in different cell types. Three highly homologous Akt isoforms have non-overlapping and opposing functions in different cancer types. As Akt is the central signalling node that incorporates cell membrane, cytoplasmic and nuclear signals regulating cell fate, analysing Akt isoforms and cell-type-specific signalling pathways and targeting them will contribute to personalised targeted HNSCC therapy. Thus, carefully designing a clinical study using a combination of a PI3K-Akt pathway inhibitor and another signalling molecule inhibitor or receptor inhibitor during the early stages of HNSCC might result in an expected positive outcome.

## Figures and Tables

**Figure 1 biomedicines-11-03001-f001:**
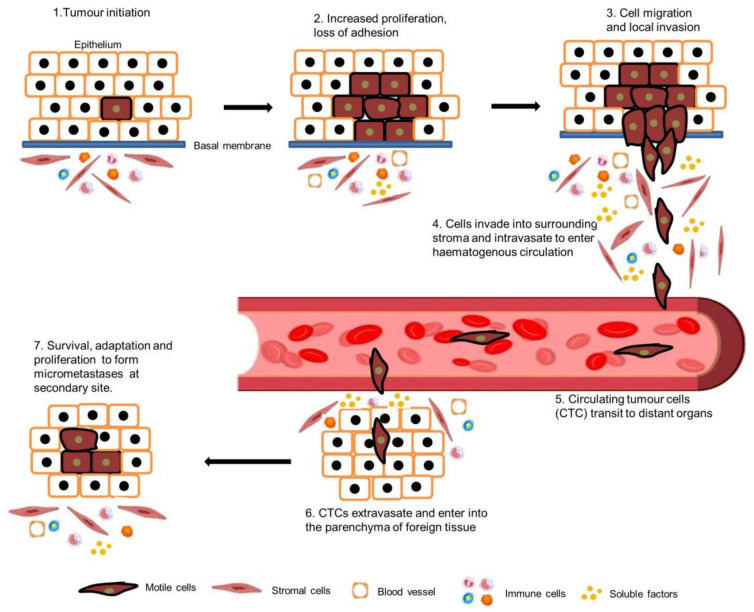
Metastasis cascade. Tumour cells proliferate uncontrollably and eventually lose their adhesive phenotype. Tumour cells then migrate and invade into surrounding tissues induced by the tumour microenvironment and intravasate to lymphatic and blood vessels. Circulating tumour cells then extravasate, enter into another tissue, and form micro-metastases at the secondary site.

**Figure 2 biomedicines-11-03001-f002:**
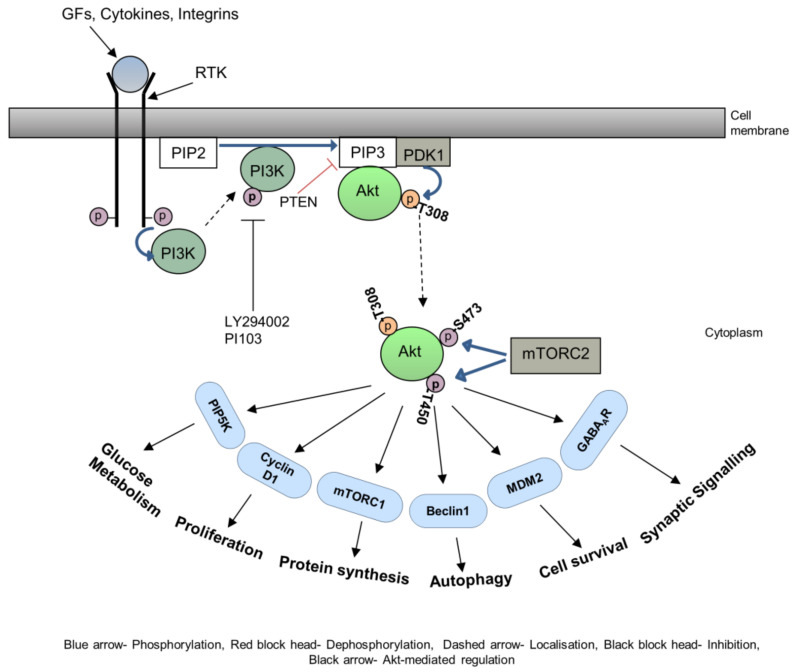
PI3K-Akt signalling pathway. Upon ligand binding, conformational changes occur in the receptor tyrosine kinase (RTK), the PI3 kinases are then activated by RTK and translocate to the plasma membrane. Activated PI3K then converts PIP2 to PIP3. Pleckstrin homology (PH) domain containing protein, Akt then translocate to the membrane, bind to PIP3, and phosphorylate at the Threonine 308 residue by PDK1. Akt translocates back to the cytoplasm and is phosphorylated further at Serine 473 and Threonine 450 residues by mTORC2. Activated Akt is responsible for initiating various cellular activities such as proliferation, protein synthesis, autophagy, cell survival, etc.

**Figure 3 biomedicines-11-03001-f003:**
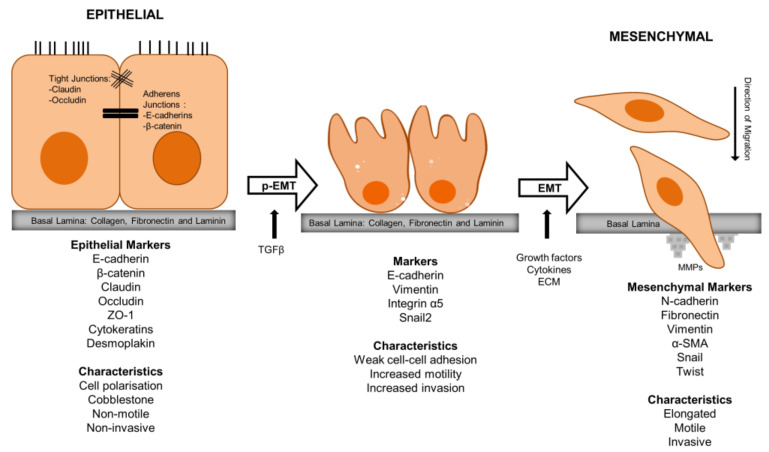
Partial epithelial to mesenchymal transition (p-EMT) and full EMT with associated biological markers. When appropriate signalling pathways are switched on, non-polarised, cobblestone-shaped epithelial cells lose their cell–cell contacts and change to mesenchymal-type motile cells. Extracellular matrix degradation enzymes, MMPs, then degrade the ECM and cells migrate through the basal lamina. This event can be detected at a molecular level by a reduction in levels of epithelial markers such as E-cadherin, b-catenin, cytokeratin, etc., and higher levels of mesenchymal markers such as vimentin, Snail, and Twist, etc. p-EMT cells also exists in a meta-stable, intermediary state between the epithelial and mesenchymal poles, suggesting this as a spectrum instead of a switch.

**Figure 4 biomedicines-11-03001-f004:**
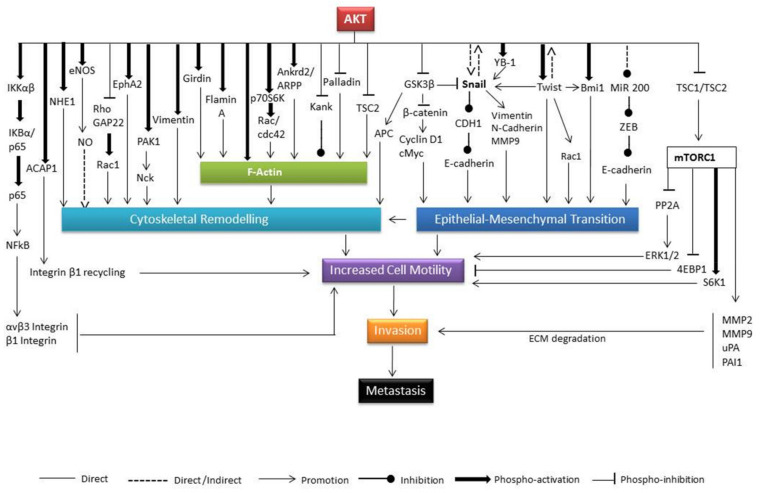
The role of Akt in the metastasis process. Cellular motility or migration is the first step of the tumour cell metastatic process. Cytoskeletal remodelling and/or epithelial to mesenchymal transition are the two cellular events that are responsible for cell migration. Akt plays significant roles in cellular migration by controlling various downstream substrates which regulate these two events. The function of Akt has been found to be cell type, tumour type, and site dependent.

**Table 1 biomedicines-11-03001-t001:** Recently completed clinical trials of PI3k/Akt pathway inhibitors in various metastatic cancers.

Trial Identifier	Phase	Type of Cancer	PI3K/Akt Inhibitor	Combination	Result	Ref
NCT01349933	II	IV/recurrent NPC	MK2206 (Akt inhibitor)	None	CR—0%, PR—4.8%, stable disease 52.4%, OS—10 months, PFS—3.5 months	[186]
NCT01370070	II	Recurrent NPC	MK2206	None	CR—0%, PR—5%, Stable disease—52%, OS—10 months, PFS—3.5 months	[187]
NCT01604772	II	IV/recurrent ADCC	MK2206	None	CR/PR—0%, Stable disease—81%, PFS—9.7 months, OS—18 months	[188]
NCT02145312	II	Recurrent/metastatic HNSCC	BYL719/Alpelisib (PI3K inhibitor)	None	Not published	[189]
NCT01527877	II	Recurrent/metastatic HNSCC	BKM120/Buparlisib (PI3K inhibitor)	None	RR—3%, Stable disease—49%, PFS—63 days, OS—143 days	[190]
NCT02021751	Ib	Recurrent/metastatic HNSCC	BYL719	Paclitaxel	Challenging safety profile, dose expansion phase was not initiated	[191]
NCT02113878	Ib	Locally advanced HNSCC	BKM120	Cisplatin/RT	Not published yet	[192]
NCT03292250	II	HNSCC	BYL719	Poziotinib (EGFR inhibitor)	Not published yet	[193]
NCT01816984	I/II	Recurrent/metastatic HNSCC	BKM120	Cetuximab	OS—9.3 months, PFS—2 months, RR—8–9%	[194]
NCT01562275	Ib	Locally advanced/metastatic solid tumours	GDC0068 (Ipatasertib)	GDC0973 (MEK1 inhibitor)	PFS—not measured due to very few participants with measurable response	[195]
NCT01625286	II	Advanced/metastatic breast cancer	AZD5363 (Capivasertib)	Paclitaxel	Adding capivasertib did not prolong PFS in the overall population	[196]
NCT01625286	II	Advanced/metastatic breast cancer	Ipatasertib	Paclitaxel	Not completed due to high number of patient death	[197]
NCT01231919	I	Recurrent solid tumours and leukaemia	MK2206		Not published	[198]
NCT01658943	II	Metastatic pancreatic cancer	MK2206	Selumetinib	OS—3.9 months, PFS—1.9 months, Disease increased—19%	[199]
NCT01971515	I	Advanced malignancy	MSC2363318A (p70S6K/Akt inhibitor)	Trastuzumab	Not published	[200]
NCT1344031	I	Postmenopausal women with metastatic breast cancer	MK2206	Anastrozole, Fulvestrant	Not published	[201]
NCT01802320	II	Recurrent/metastatic colon cancer	MK2206		OS—6.8 months, PFS—1.8 month, ORR—O%	[202]
NCT01245205	I	Metastatic solid tumour/breast cancer	MK2206	Lapatinib	Not published	[203]
NCT01783171	I	Pancreatic cancer	MK2206	Dinaciclib	Not published	[204]
NCT01333475	Pilot	Advanced colorectal cancer	MK2206	Selumetinib	Biomarker analysis	[205]
NCT01979523	II	Metastatic uveal melanoma	GSK2141795	Trametinib	PFS—15.6 wks, OS—88 wks, disease progressed—22%	[206]
NCT01604772	II	Adenoid cyst carcinoma	MK2206		PFS—9 months, PS—18 months, Grade 3 adverse event—62%	[207]
NCT01896531	II	Metastatic gastro-oesophageal cancer	Ipatasertib	mFOLFOX6	pFS—6.57 months, OS—12 months, ORR—52%	[208]
NCT01349933	II	Metastatic HNSCC	MK2206		Disease progressed or dead: 57%, PR—4.8%	[186]
NCT01263145	I	Metastatic solid or breast cancer	MK2206	Paclitaxel	Not published	[209]
NCT01964924	II	Metastatic triple-negative breast cancer	GSK2141795	Trametinib	ORR—5.4%, CBR—31.2%	[210]
NCT02018874	Ib	Solid tumours and non-Hodgkin’s Lymphoma	LY2780301(p70S6K/Akt inhibitor)	Gemcitabine	Not published	[211]

RR = Response rate, ORR—objective response rate, CR—complete response, PR—partial response, CBR = CR + PR + stable disease, OS—overall survival, PFS—progression-free survival, NPC—nasopharyngeal cancer, ADCC—adenoid cystic carcinoma.

## Data Availability

No new data were created or analysed in this study. Data sharing is not applicable to this article.

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
