# Peer review of "Role of Akt/Protein Kinase B in Cancer Metastasis"

_biomedicines, 2023, doi:10.3390/biomedicines11113001_

Round 1

Reviewer 1 Report

Comments and Suggestions for Authors

This review focus on the role of Akt in the regulation of invasion and metastasis in various cancers including head and neck squamous cell carcinoma (HNSCC). Overall, this review is well-organized. Several points should be noted as below.

1) It needs more updates about cancer metastasis, not only that “Metastasis is a critical step in the process of carcinogenesis”, etc. One recent paper proposes that cancer is a multidimensional spatiotemporal "unity of ecology and evolution" pathological ecosystem, and cancer metastasis includes NPC should be a multidirectional ecological dispersal (https://www.thno.org/v13p1607.htm). It is interesting to add something new.

2) Figure 1, the motile cells is EMT cells?

3) Please talk something about partial-EMT in this paper and Figure 3 might be better revised accordingly.

4) How about the combination of PI3k/Akt inhibitors and immunotherapy in HNSCC?

Reviewer 2 Report

Comments and Suggestions for Authors

In the review manuscript titled “Role of Akt/Protein Kinase B in Cancer Metastasis”, Islam et al. present a comprehensive summary of recent advancements in understanding the role of Akt in regulating tumor cell migration and invasion. They also discuss the status of PI3K-Akt pathway inhibitors in clinical trials for neck squamous cell carcinoma. This paper serves as an excellent resource for those outside the field and the undergraduate readers, offering a clear overview of foundational knowledge. This review paper seems to miss the depth expected of a comprehensive review. Instead of delving into detailed discussions and providing insights into the subject matter, it appears to merely enumerate knowledge points. A more in-depth analysis or synthesis of the presented information would have greatly enhanced the value of the paper. As it stands, readers might struggle to grasp the broader implications and context of the listed points. It would be beneficial if the authors revisited their approach to offer a more holistic overview of the topic.

1.The relevance of Figures 1, 2, and 3 seems limited. It would enhance the manuscript if the authors could more directly tie these figures to the topics discussed.

2.Why did the authors specifically focus on neck squamous cell carcinoma and not other cancers? For a review paper, a comprehensive summary of inhibitors targeting Akt across a spectrum of cancers would have been beneficial for readers.

3.The font size and type used in the figures should be uniform to enhance their readability. Typically, 'Times New Roman' or 'Arial' are preferred font types

Comments on the Quality of English Language

Minor editing of English language required

Round 2

Reviewer 1 Report

Comments and Suggestions for Authors

The authors have completely answered my questions. 

Reviewer 2 Report

Comments and Suggestions for Authors

Thank you to the authors for their response; all of my concerns and comments have been addressed.